# Predictive Maintenance for Optical Networks in Robust Collaborative Learning

## Abstract

Machine learning (ML) has recently emerged as a powerful tool to enhance the proactive optical network maintenance and thereby, improve network reliability and operational efficiency, and reduce unplanned downtime and maintenance costs. However, it is challenging to develop an accurate and reliable ML based prognostic models due mainly to the unavailability of sufficient amount of training data since the device failure does not occur often in optical networks. Federated learning (FL) is a promising candidate to tackle the aforementioned challenge by enabling the development of a global ML model using datasets owned by many vendors without revealing their business-confidential data. While FL greatly enhances the data privacy, a global model can be strongly affected by a malicious local model. We propose a robust collaborative learning framework for predictive maintenance on cross-vendor in a dishonest setting. Our experiments confirm that a global ML model can be accurately built with sensitive datasets in federated learning even when a subset of vendors behave dishonestly.

## 1 Introduction

Optical fiber networks compose the core of telecommunication infrastructure today due to their high capacity of data transmission. Optical networks rely on fully functional hardware components that run under optimal conditions. In order to reduce the risk of unplanned network interruption and service outage, it is important to predict the degradation of hardware network components correctly using analyzing tools and techniques, by which the maintenance budget and resources are allocated efficiently and timely. Due to the great benefits in industry, global predictive maintenance market is expected to reach more than $13 billion by 2026 (ReportLinker, 2021; Simon, 2021).

Machine learning (ML) based prediction is an emerging method to improve the accuracy of predictive maintenance in manufacturing industry and communication networks. An ML model is trained by the historical data of hardware failure and then the upcoming maintenance is predicted by real-time data gathered through measurement at the edge. ML techniques can be useful if a sufficiently large, diverse, and realistic set of training data exists. Since an ML model relies so heavily on good training data, the availability of such datasets is a crucial requirement for this approach.

However, it is challenging to develop a high-precision ML model for predictive maintenance mainly due to the lack of training data. The hardware failures or maintenance events do not occur frequently so that it takes time until good and meaningful training data are collected through the network. Hence, an ML model is often trained using the accelerated aging test results (e.g. a life cycle under the extreme temperature or the over-powered condition) that are conducted by hardware manufacturers. Since the components of network equipment are usually produced by small and medium-sized companies, such an ML model is trained based on the limited amount of data that are owned by each manufacturer.

This situation can be relieved, if the training dataset can be aggregated from multiple vendors and consolidated in a central location. Since collaborative learning allows to train a model on larger datasets rather than the dataset available in a single vendor, a higher quality and more accurate ML model can be built. However, such collaboration is not straightforward in reality since vendors are not willing to share their training data with external companies. Aging test data are often company-confidential and trade secret. Moreover, sharing data with foreign companies may be prohibited by privacy protection regulations in their home countries.

**Federated Learning**  Federated learning (FL) is a framework of enabling distributed parties to work together to train machine learning models without sharing the underlying data or trusting any of the individual participants (Bonawitz et al., 2017b). FL can be used to build an ML model from various companies for the purpose of predicting the failures, repairs, or maintenance of network systems. With the FL technique, the training data is not required to be centralized, but can instead remains with the data owners. Each vendor trains an ML model on their private data and using their own hardware. These models are then aggregated by a central server (e.g. a network operator) to build a unified global model that has learned from the private data of every vendor without ever directly accessing it. Hence, confidential training data (e.g. aging test results of products) are not visible to a server, nor other competitive vendors.

**Secure aggregation**  Secure aggregation in FL is a cryptographic protocol that enables each vendor to submit a local model securely and a server learns nothing but the sum of the local models. A secure aggregation method for mobile networks was presented in Bonawitz et al. (2017b) and Bell et al. (2020). This method relies on a pairwise secret exchange and Shamir's $t$-out-of-$n$ secret sharing scheme, focusing on the setting of mobile devices where communication is extremely expensive, and dropouts are common.

There is a rich literature exploring secure aggregation in both the single-server setting (via additive masking Bonawitz et al. (2016), via threshold homomorphic encryption (HE) Halevi et al. (2011), and via generic secure multi-party computation (MPC) Burkhart et al. (2010)) as well as in the multiple non-colluding servers setting (Corrigan-Gibbs & Boneh, 2017). For instance, one can perform all computations using a fully homomorphic encryption scheme resulting in low communication but very high computation, or using classical MPC techniques with more communication but less computation. Other works use a hybrid of both and thus enjoy further improvement in performance (Juvekar et al., 2018; Mishra et al., 2020). Nevertheless, it is still an open question how to construct a secure and robust aggregation protocol that addresses all the challenges.

**Our contribution**  In this paper, we propose a secure and robust collaborative learning method using cross-vendor datasets for predictive maintenance in optical networks. Each vendor builds a local model using its own training dataset and uploads it to the server. The private dataset remains in the vendor's domain and is never exposed to other companies. A server builds a global ML model by aggregating local ML models iteratively and averaging them to form an updated global model proportional to the size of dataset. Using the global model, the potential risk of hardware failure and corresponding maintenance events are predicted and the necessary resources are proactively prepared to run optical networks without disruption.

In our framework, a secure aggregation protocol is tolerant to the malicious behavior of participants in a honest-majority model; that is, a server and majority of vendors are assumed to be honest yet some may be malicious or unreliable. Compared to the original FL, the local models are not many, and the dropouts are very rare in our framework. Furthermore, an updated global model is not shared with vendors. The reason is that, while a global model is a valuable asset to the network management, it is not really beneficial to the vendors. Instead, each vendor receives the personalized maintenance report which contains the discrepancy between its local model and the global model, which is useful to improve the quality of products in the future. Fig. 1 shows that an example of the ML-based predictive maintenance process in FL under the assumption that a single vendor behaves maliciously.

**Related work**  In Bonawitz et al. (2017a), a practical secure aggregation technique in an FL setting was proposed over large mobile networks. Such framework does not fit for our use case due to multiple reasons. Firstly, in our use case, a global model is not shared with data owners (vendors). Each vendor gets benefit by receiving an individual maintenance result (e.g. the difference between the prediction and the real failure) after the global model is deployed and hardware degradation is predicted. Secondly, the scalability is not important since the number of vendors are not very large and dropouts are expected to be rare. On the other hand, secure aggregation is critical since the disclosure of the private training dataset may give negative impact on the data owner's business.

Another interesting work on collaborative predictive maintenance was presented in Mohr et al. (2021), where a combination of blockchain and federated learning techniques was applied. We apply a multi-party computation technique for data privacy since it is more suitable for our use case.

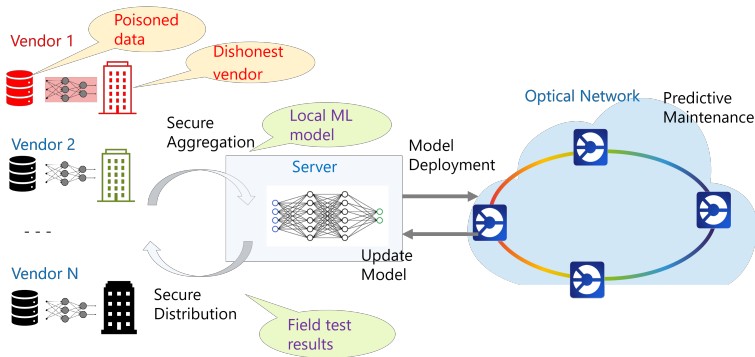

Figure 1: ML-based predictive maintenance process in a dishonest setting

More recently, in Zheng et al. (2021), an end-to-end platform for collaborative learning using MPC is proposed. Though it is an interesting approach, it is unlikely that this platform can be applied to our use case since the collaborative learning through the use of release policies and auditing is not preferable to the predictive maintenance.

The rest of this paper is structured as follows: The use case and methods of ML-based predictive maintenance are presented. Then, the threat scenarios and defending methods are presented. Our experimental results are presented and the conclusion is given.

## 2 ML-BASED PREDICTIVE MAINTENANCE

### 2.1 USE CASE: OPTICAL TRANSMITTER DEGRADATION PREDICTION

Semiconductor lasers are considered as one of the most commonly used optical transmitters for optical communication system thanks to their high efficiency, low cost, and compactness. They have been rapidly evolved to meet the requirements of the next generation optical network in terms of high speed, low power consumption etc. However, during operation the performance of the laser can be adversely impacted by several factors such as contamination, facet oxidation etc. Such factors are hard to predict and cause the laser degradation and failure, and thereby resulting in optical network disruption and high maintenance costs. Therefore, it is highly required to predict the degradation of the semiconductor laser device after its deployment in optical communication system in order to enhance the system reliability and minimize the downtime costs.

ML techniques could provide a great potential to tackle the laser degradation prediction problem (Abdelli et al., 2020). The development of such prognostic methods requires the availability of run-to-failure data sets modelling both the normal operation behavior and the degradation process under different operating conditions. However, such data is often unavailable due the scarcity of the failures during the system operation and the long time required to monitor the device up failing and then generating the reliability data. That's why accelerated aging tests are often adopted to collect run-to-failure data in shorter time by speeding up the device degradation by applying accelerated stress conditions resulting in the same degradation process leading to failure (Celaya et al., 2011).

However, the burn-in aging tests are carried out just for few devices due to the high costs of performing such tests. Hence, the amount of the run-to-failure data that can be derived from such tests, might be small, which can adversely affect the performance of ML model (Abdelli et al., 2021). Therefore, a FL approach is considered as a promising candidate to address the aforementioned problem, whereby different semiconductor laser manufacturers (i.e vendors) collaborate with their small local dataset, stored at their premise, in order to build an accurate and reliable global laser degradation prediction model with good generalization and robustness capabilities.

Note that the semiconductor laser manufacturers might have different types of laser devices with various characteristics leading to different degradation trends, and that the data owned by each vendor is derived from aging tests conducted under different operating conditions. State that the global

model is run in a server hosted by an optical network operator owning the infrastructure in which the semiconductor lasers manufactured by the different vendors are deployed.

We consider a FL system composed of a server and $N$ clients (i.e vendors) that collaboratively train a global model using the FedAvg algorithm (McMahan et al., 2017). The clients send securely the local model weight updates to the server using MPC. The autoencoder based on gated recurrent unit (GRU) is used as global model to solve the task of semiconductor laser degradation prediction. A convolutional autoencoder implemented at the server is adopted as an anomaly detection method to detect the anomalous weights sent by the malicious clients.

## 2.2 METHODS

**Autoencoder** An autoencoder (AE) is a type of artificial neural network seeking to learn a compressed representation of an input in an unsupervised manner (Kramer, 1991). An AE is composed of two sub-models namely the encoder and the decoder. The encoder is used to compress an input $X$ into lower-dimensional encoding (i.e. latent-space representation) $Z$ through a non-linear transformation, which is expressed as follows:

$$Z = f(WX + b), \tag{1}$$

where $W$ and $b$ denote the weight and bias matrices of the encoder and $f$ represents the activation function of the encoder.

The decoder attempts to reconstruct the output $\hat{X}$ given the representation $Z$ via a nonlinear transformation, which it is formulated as follows:

$$\hat{X} = g(W'X + b'), \tag{2}$$

where $W'$ and $b'$ represent the weight and the bias matrices of the decoder and $g$ denotes the activation function of the decoder.

The AE is trained by minimizing the reconstruction error between the output $\hat{X}$ and the input $X$, which is the loss function $L(\theta)$, typically the mean square error (MSE), defined as:

$$L(\theta) = \sum ||X - \hat{X}||^2 \tag{3}$$

where $\theta = \{W, b, W', b'\}$ denotes the set of the parameters to be optimized.

**Gated Recurrent Unit (GRU)** GRU recently proposed by (Cho et al., 2014) to solve the gradient vanishing problem Cho et al. (2014), is an improved version of standard recurrent neural networks (RNNs), used to process sequential data and to capture long-term dependencies. The typical structure of GRU contains two gates namely reset and update gates, controlling the flow of the information. The update gate regulates the information that flows into the memory, while the reset gate controls the information flowing out the memory. The GRU cell is updated at each time step $t$ by applying the following equations:

$$z_t = \sigma(W_z \cdot x_t + W_z \cdot h_{(t-1)} + b_z) \tag{4}$$

$$r_t = \sigma(W_r \cdot x_t + W_r \cdot h_{(t-1)} + b_r) \tag{5}$$

$$\hat{h_t} = \tanh(W_h \cdot x_t + W_h \cdot (r_t \circ h_{(t-1)}) + b_h) \tag{6}$$

$$h_t = z_t \circ h_{(t-1)} + (1 - z_t) \circ \hat{h_t} \tag{7}$$

where $z_t$ denotes the update gate, $r_t$ represents the reset gate, $x_t$ is the input vector, $h_t$ is the output vector, $W$ and $b$ represent the weight and the bias matrices respectively. $\sigma$ is the gate activation function and $tanh$ represents the output activation function. The '·' operator denotes a matrix multiplication, the '∘' operator represents the dot product.

## 2.3 ANOMALY DETECTION

AE has been widely used for anomaly detection by adopting the reconstruction error as anomaly score. It is trained with only normal data representing the normal behavior. After training, AE will reconstruct the normal instances very well, while it will fail to reproduce the anomalous observations by yielding high reconstruction errors. The process of the classification of an instance or

observation as anomalous/normal is shown in Alg. 1. If the calculated anomaly score is higher than a set threshold $\theta$, the instance is classified as "anomalous", else it is assigned as "normal". $\theta$ is a hyperparameter optimized based on the number of true positives and false positives.

---

**Algorithm 1** Autoencoder-based anomaly detection algorithm

---

**Input** Normal dataset $X$, Anomalous dataset $x^{(i)}, i = 1, \ldots, N$, threshold $\theta$
**Output** Reconstruction error $||x - \hat{x}||$
$f, g \leftarrow$ train an autoencoder using the normal dataset $X$
**for** $i = 1, \cdots, N$ **do**
    reconstruction error(i) $\leftarrow ||x^{(i)} - g \circ f(x^{(i)})||$
    **if** reconstruction error(i) $> \theta$ **then**
        $x^{(i)}$ is anomalous.
    **else**
        $x^{(i)}$ is normal.
    **end if**
**end for**

---

## 3 ROBUST COLLABORATIVE LEARNING

We consider training an ML model in a federated learning setting, wherein each vendor maintains a private dataset of its own hardware. A global ML model is trained under the coordination of a central server based upon multiple local models that are provided by different vendors. A server can get only a sum of the local models and does not see local models individually. Based on the global model, the maintenance events in optical networks are predicted and the corresponding materials are prepared accordingly.

### 3.1 THREATS

An important challenge in FL is to prevent a server or other vendors from reconstructing the private data of any vendor while collaborating at any circumstances. While a secure aggregation protocol in FL addresses the strong privacy of the data of the vendors, the FL framework creates a new attack surface during the model training process. Since the vendors have full control over local training processes, they may submit arbitrary updates to change the global model without being detected. Among the broad range of attacks on FL, following attacks are most relevant to our use case.

**Model inversion attack** An attacker can intercept the updated local models and extract the private training data from the models. For example, in Fredrikson et al. (2015), authors demonstrated a model inversion attack that could extract images from a face recognition system, which look suspiciously similar to images from the underlying training data. The model inversion attacks can be mitigated by applying differential privacy techniques; adding noise to the local models before sending them to the server. However, such noise will degrade the overall model performance, which is not preferable to our use case.

**Local model poisoning attack** This attack injects poisoned instances into the training data, or directly manipulates model updates during the aggregation protocol. Instead of locally trained models, these Byzantine vendors may upload the poisoned local models which are highly deviated from the global model. As a result, the attacker can tamper with the weights of the global model or inject a backdoor into it, misclassifying specific inputs into the target class as intended by the attacker.

**Active corruption** A secure aggregation protocol enables each vendor to submit a local model securely and the server learns nothing but the sum of the local models. In this attack, some corrupted vendors may arbitrarily deviate from the secure aggregation protocol. Some vendors do not follow the protocol honestly and provide wrong values to the server or other vendors.

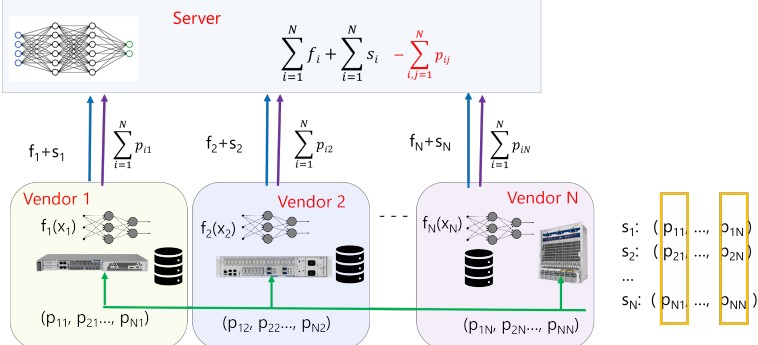

Figure 2: Secure collaborative learning using Secret Sharing in FL

## 3.2 DEFENSES

We present two defending methods against attacks in FL: MPC-based secure aggregation against the model inversion attack and anomaly detection against the model poisoning attack and the active corruption attack.

**MPC-based Secure aggregation** Suppose that the server and vendors (clients) behave honestly but curiously (semi-honest model). That is, all participants follow the protocol exactly as instructed but also try to retrieve the private data of other vendors, if possible. Under this scenario, a simple $n$-out-of-$n$ additive secret sharing scheme can be used to prevent the model inversion attack as well as keep the privacy of local models.

Suppose $N$ is the number of clients and each client has its own local model $f_i$ where $1 \leq i \leq N$. The client $i$ generates a random linear mask $s_i$ and sends $f_i + s_i$ to the server. Also, the client $i$ divides $s_i$ into $N$ additive shares, $\{p_{i1}, \ldots, p_{iN}\}$, in such a way that $s_i = \sum_{j=1}^{N} p_{ij}$. Note the size of $s_i$ is similar to those of shares. These $N$ shares are distributed to other clients in such a way that each client receives a unique share out of $N$ shares. In result, the client $i$ receives $\{p_{1i}, \ldots, p_{Ni}\}$. Finally, the client $i$ sends the sum of the shares $\sum_{j=1}^{n} p_{ji}$ to the server. This process is repeated for all clients.

By aggregating one-time padded local models and the sum of the shares, the server can calculate the sum of the local models as follows:

$$\sum_{i=1}^{N}(f_i + s_i) - \sum_{i=1}^{N}\sum_{j=1}^{N} p_{ji} = \sum_{i=1}^{N} f_i + \sum_{i=1}^{N}\left(s_i - \sum_{j=1}^{N} p_{ij}\right) = \sum_{i=1}^{N} f_i \tag{8}$$

A pseudo code of the secure aggregation protocol is given in Appendix A.1. An overview of the secure collaborative learning procedure is shown in Fig. 2.

**Autoencoder based anomaly detection** The autoencoder is trained with a dataset $D = \{w^1, w^2 \ldots w^N\}$ incorporating the model weights sent by trusted clients (i.e normal weights) and stored at the server. The dimensionality of the model weight $w^k$ is reduced to produce a low-dimensional input in order to reduce the computational complexity due to the high dimension of the model weight. The generated input is fed then to the autoencoder for training, whereby the encoder compresses the input into a lower-dimensional latent vector which is then reconstructed by the decoder. After the training phase, the autoencoder is able to recognize the normal weights and mark any weight that deviates from the data seen during the training as an anomaly. The reconstruction error between the input weight and the reconstructed weight is used as an anomaly score. If the anomaly score exceeds a pre-defined threshold, the weight is recognized as anomalous potentially sent by a malicious client, and thereby it is removed and not considered for the update of the global model. The threshold is optimized in order to improve the detection capability of the autoencoder for different poisoning model attacks.

## 4 EXPERIMENTS

### 4.1 EXPERIMENTAL DATA

The experimental data is derived from various accelerated aging tests performed for different semiconductor laser devices operating under several operating conditions and carried out under high temperature ( $\geq 50°C$ ) to strongly increase the laser degradation and thereby speed up the device failure. The output power (i.e degradation parameter) is monitored under constant current, until 15,000 h. The failure or degradation criteria of the device is defined as the decrease of the output power by 20% of its initial value. Few devices are degraded or failed either during the aging test or after the end of the test. In total, a dataset of 6,786 samples incorporating the sequences of monitored output measurements is built. We assign to each sample the state of the device (normal or degraded (i.e anomalous ) by applying the aforementioned failure criteria. The said data is then normalized and randomly divided into a training data (comprising of 80% of the samples) and a test dataset (the remaining 20% for testing). The training dataset incorporates only samples of normal devices, whereas the test dataset includes samples of both normal and degraded devices. The training data is split then into N=10 clients with different parts of 450, 500, 554, 700, 382, 520, 450, 445, 300, and 380 respectively, leading to heterogeneous federated setting.

### 4.2 GLOBAL MODEL

Given that the data is highly unbalanced due to the small number of failed devices, adopting supervised methods can be unfeasible due to the lack of adequate number of normal and fault data. Therefore, an unsupervised method namely the GRU-based autoencoder (GRU-AE) is used as global model. GRU is adopted to capture the laser degradation trend in sequential input. The GRU-AE is trained with normal data underlying the normal behavior of the laser device, and tested with observations of normal and failed devices. Note that the input sequences fed to the GRU-AE include the output power measurements collected only till 5,000 h in order to train the model to early predict the degradation and that it is tested with sequences of normal and degraded devices failed after 5,000 h to evaluate the early prediction capability. The input of the global model consists of a 6-length sequence of historical output power measurements combined with the operating conditions features namely the temperature and the current, impacting the degradation trend. The architecture of global model is composed of two GRU layers containing each 64 cells. Rectified Linear Unit (ReLU) is selected as an activation function for the hidden layers of the model. The training of the global model is carried out in an iterative process as follows:

- The server distributes the global model $w_t^G$ to N clients.
- Each client $k$ trains the model locally using its local data $D_k$, and updates the weight $w_t^k$ for $\alpha$ epochs of Adam with mini-batch size of $\beta$ to compute $w_{(t+1)}^k$.
- The server securely aggregates each client's $w_{(t+1)}^k$ using MPC.
- An autoencoder-based anomaly detection method is used to detect anomalous weights sent by the clients.
- The update of the global model $w_{(t+1)}^G$ is computed by a weighted averaging of only normal weights.

The above-described process is repeated for multiple communication rounds $N_{round}$ (e.g. number of aggregation) to improve the performance of the global model. For our experiments, $\alpha$, $\beta$ and $N_{round}$ are set to 10, 16 and 100 respectively.

### 4.3 ANOMALOUS WEIGHT DETECTION METHOD

A convolutional autoencoder implemented at the server is used to identify the anomalous weights and thereby detect the potentially malicious clients. The model contains an encoder and a decoder sub-model with 5 layers. The encoder takes as an input a vector of length 64. It encodes the input into low dimensional features through a series of 2 convolutional layers containing 64 and 32 filters of size $3 \times 1$ with a stride of 2 and 1, respectively. The decoder is inversely symmetric to the encoder part. It consists of 3 transposed convolutional layers used to up-sample the feature maps. The last

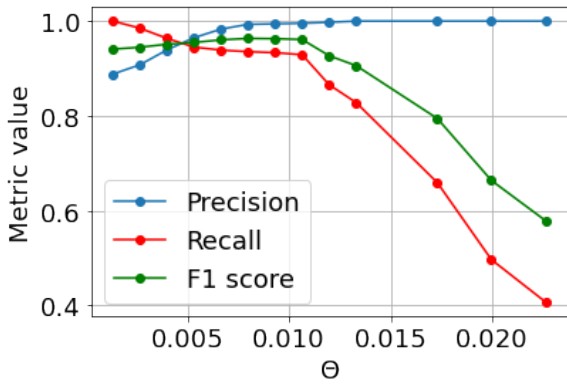

Figure 3: The optimal threshold selection based on the precision, recall and F1 score scores yielded by the GRU-based autoencoder

transposed convolutional layer with 1 filter of size $3 \times 1$ and a stride of 1 is used to generate the output. ReLU is selected as an activation function for the hidden layers of the model. The loss function is set to the MSE, which is adjusted by using the Adam optimizer.

### 4.4 EXPERIMENTAL RESULTS

**Prediction Capability Evaluation**   We evaluate the traditional centralized approach by applying the GRU-AE to a centralized aggregated data containing the datasets of all the clients. The degradation prediction capability of the GRU-AE is optimized by selecting an optimal threshold $\theta$ ensuring the best precision and recall tradeoff. Figure 3 illustrates the precision, recall and F1 score curves along with $\theta$. If the chosen threshold is too high, many degraded laser devices will be classified as normal devices, resulting in higher false positive. Whereas if the selected threshold is too low, many normal devices will be classified as abnormal, leading to higher false negative. Therefore, the optimal threshold that maximizes F1 score is chosen. The same selected threshold to distinguish the normal devices from the degraded ones is used for the FL approach.

The performance of the FL approach is compared to the centralized approach using as evaluation metrics precision, recall, F1 score and accuracy. The results of the comparison shown in Table 1 prove that the FL framework achieves similar performance as the centralized approach.

Table 1: Comparison of FL and centralized approach

| Approach | Precision (%) | Recall (%) | F1 score(%) | Accuracy (%) |
|---|---|---|---|---|
| FL | 99.31 | 93.22 | 96.17 | 93.42 |
| Centralized | 99.31 | 93.54 | 96.34 | 93.71 |

**Attack detection capability evaluation**   The anomalous weight detection model is compared to defense-based methods namely krum (Blanchard et al., 2017), Trimmed Mean (Yin et al., 2018) and Median. The reconstruction error achieved by the global model for each global round is adopted as evaluation metric. Two adversarial attacks namely additive noise (i.e adding gaussian noise to the model weight) and sign flip (i.e flipping the sign of the model weight) (Li et al., 2018) are generated by 10% of clients for each round. The results shown in Figure 4 demonstrate that the proposed method significantly outperforms the defense-based approaches for the considered attack scenarios. It can be seen that the proposed method achieves similar performance as the FedAvg algorithm without attack, which proves the effectiveness of the anomaly detection model in detecting the anomalous weights. The performances of the defense-based methods are worse as they are not effective in defending against attacks for not identically and independently distributed (iid) settings,

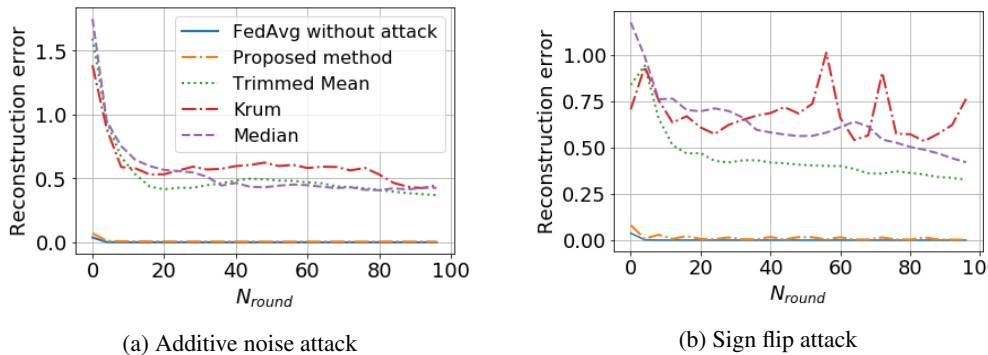

(a) Additive noise attack                     (b) Sign flip attack

Figure 4: Comparison of the proposed detection-based method and defense-based approaches in terms of reconstruct error of the global model under two adversarial attacks. Note that the legend in Fig. 4a applies to Fig. 4b as well.

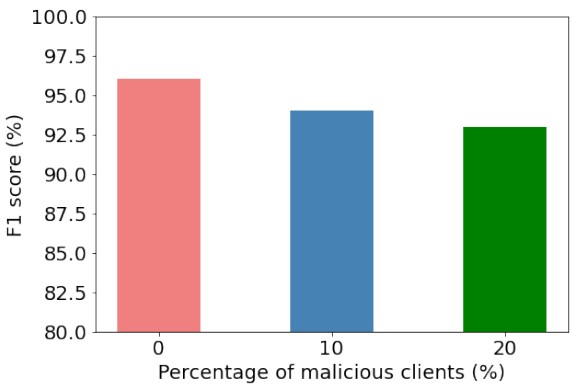

Figure 5: Evaluation of the performance

and the fraction of the malicious clients which is required by Krum and Trimmed Mean can not be known a priori in FL.

We evaluate the performance of the global model in terms of F1 score by increasing the percentage of malicious clients to 20%. The results depicted in Fig. 5 show that the model can still achieve good performance even if the 20% of the clients are malicious, which proves that the anomaly detection method is able to recognize the anomalous weights. Note that the training data for FL approach is not big (4200 samples) and if 10% or 20% of clients are malicious, and the anomaly detection method accurately detect them and remove the anomalous weights, the global model is trained in such cases with fewer clients, thus fewer data, which might impact the performance of the model.

## 5 CONCLUSION

Optical networks require a high level of reliability and sustainability. Machine learning techniques are expected to improve maintaining such networks efficiently. We showed that an accurate and reliable ML model could be developed in collaborative learning without the disclosure of clients' sensitive datasets even in a malicious setting. Our experiments confirm that (i) the presented FL approach achieves a good prediction capability similar to the one yielded by the centralized approach, and (ii) the proposed autoencoder based anomaly detection model is efficient in recognizing the anomalous weights potentially sent by malicious clients, and outperforms the defense-based methods.

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

## A APPENDIX

### A.1 AGGREGATION PROTOCOL

Suppose that a server (network provider) builds a global ML model for predictive maintenance with $N$ vendors. The pseudo code of the algorithm is given in Algorithm 2.

---

**Algorithm 2** Federated averaging algorithm using MPC

---

**Input** The $N$ vendors are indexed by $i$; $x_i$ is the local training dataset; $s_i$ is the linear mask; $p_{ij}$ is the $j$-th share of the linear mask $s_i$;
**Output** A global ML model $F$

**for** round $t = 1, 2, \cdots$ **do**
    **for** $i = 1, 2, \ldots, N$ **do**
        $f_i^{t+1} + s_i^{t+1} \leftarrow LocalUpdate(i, f_i^t)$
        $q_i^{t+1} \leftarrow LocalShares(i)$
    **end for**
    $F^{t+1} \leftarrow \sum_{i=1}^N (f_i^{t+1} + s_i^{t+1} + q_i^{t+1}) = \sum_{i=1}^N f_i^{t+1} + \sum_{i=1}^N (s_i^{t+1} + \sum_{j=1}^N p_{ij}^{t+1}) = \sum_{i=1}^N f_i^{t+1}$
**end for**

$LocalUpdate(i, f)$:
$\mathcal{B} \leftarrow$ (split $x_i$ into batches of size $B$)
**for** each epoch $B$ **do**
    **for** batch $b \in \mathcal{B}$ **do**
        $f \leftarrow f - \eta \cdot A(f, b)$
    **end for**
    $f \leftarrow f + s_i$
**end for**
return $f_i$ to the server.

$LocalShares(i)$:
**for** $j = 1, 2, \ldots, N$ **do**
    $q_i \leftarrow \sum_{j=1}^N p_{ij}$
**end for**
return $q_i$ to the server.

---

