# OpenReview forum: "Predictive Maintenance for Optical Networks in Robust Collaborative Learning  "
_ICLR.cc/2022/Conference — ICLR 2022 Submitted_

### Official Review · Reviewer_qcjG · 2021-10-21

**Correctness:** 2
**Technical Novelty And Significance:** 2
**Empirical Novelty And Significance:** 2
**Recommendation:** 3
**Confidence:** 4

**Main Review:**

•	The main challenge in predictive maintenance is often models trained on a specific unit/asset will not generalize and perform well when tested on a different unit of the same type. For example, if the model is trained on a ball bearing under certain operating condition and then tested on a different bearing under a different operating condition, the result will not be good. In the proposed concept of collaborative learning with different datasets from different vendors, it is very unlikely that the different vendors use relatively similar assets. The paper data set is not from different vendors therefore the prediction of the model under such condition will be not acceptable.  This challenge is not addressed in this paper.

•	The paper should state exactly whether it is tackling a prognostic (prediction of remaining useful life) or Diagnostic (detection of the fault type) or anomaly detection problem. This is not clear from the paper. In page three authors mention of prognostics, later on an AE is used for anomaly detection. Please elaborate in the paper what type of problem is being solved here.

•	The paper uses accelerated testing data, while such data is widely used to build reliability models for different assets, their application on the normal condition experiment is questionable. The model trained on accelerated life tests cannot be tested on normal condition data.

•	The description of the dataset should be enhanced in the paper. Not enough information is reported about the types of faults/degradation/experimental setup/differences of the experimental setups used

•	While the concept is based on collaborating learning of different vendors that may use very different assets for the training sets. The dataset that the authors use is not from various/ different assets. There only difference is in operating conditions. Unit to unit variation, similar device manufactured by different vendors with, is a big challenge in predictive maintenance that can not be addressed with the above dataset used in the paper. So my main point is the dataset is not suitable for demonstration of the collaborative learning concept in predictive maintenance.

•	The key contribution from the algorithm perspective is not clear in the paper. Please elaborate in the paper if there are any algorithmic contribution in the paper.

•	The paper states that a Robust and secure model is developed. However, it does not provide some clear criteria for robustness as well as security. So these aspects of the model should be developed further in details or may be in a separate manuscript/appendix.


**Summary Of The Paper:**

The paper is on a very interesting topic, predictive maintenance, that has a big impact on various industries. It presents a concept/framework for collaborative learning in predictive maintenance application where a global model is trained based on different vendors training dataset without sharing the data.

**Summary Of The Review:**

While the presented concept of collaborative learning for predictive maintenance can be a key contribution to the domain of PdM, this idea is not well developed by means of related datasets. One suggestion is, the authors use several bearing datasets that are available publicly, i.e. Case Western Reserve University dataset, University of Cincinnati’s dataset and many more available bearing datasets and use them as different vendors and then train the global model. Then the model can be tested on an unseen different dataset. The output should be compared with ground truth if the authors are solving a prognostics problem or be compared to a known fault type if they are solving a diagnostics classification problem.
With the above said enhancements needed for the paper, I cannot recommend the paper for the publication.

---

### Official Review · Reviewer_BxDo · 2021-11-03

**Correctness:** 2
**Technical Novelty And Significance:** 2
**Empirical Novelty And Significance:** 2
**Recommendation:** 3
**Confidence:** 4

**Details Of Ethics Concerns:**

None.

**Main Review:**

The focus of this study is not clear, whether it is to solve the challenge of collaborative failure prediction or the problem of encryption aggregation in federated learning (FL). In fact, the related work part only summarized three related studies. I believe this is far from enough to reflect the related research progresses. On the one hand, secure aggregation is a fundamental task of FL. We can see a lot of recent progress (see below, for example). Although, as the authors said, they typically focus on the global model's accuracy, efficiency, and scalability issue. However, it does not mean that they are not suitable for the problem scenarios of collaborative prediction of optical transmitter degradation simply because we do not need these features.

-	Fereidooni, Hossein, et al. "SAFELearn: secure aggregation for private federated learning." IEEE Security and Privacy Workshops (SPW), 2021.

-	Li, Yong, et al. "Privacy-Preserving Federated Learning Framework Based on Chained Secure Multiparty Computing." IEEE Internet of Things Journal 8.8 (2020): 6178-6186.

-	Kadhe, Swanand, et al. "Fastsecagg: Scalable secure aggregation for privacy-preserving federated learning." arXiv preprint arXiv:2009.11248 (2020).

-	Yang, Chien-Sheng, et al. "LightSecAgg: Rethinking Secure Aggregation in Federated Learning." arXiv preprint arXiv:2109.14236 (2021).

-	Constance Beguier, et al. " Efficient Sparse Secure Aggregation for Federated Learning." arXiv preprint arXiv:2007.14861 (2021).

On the other hand, FL-based prediction tasks have also been extensively studied, although they are not necessarily designed for optical network predictive maintenance.

The motivation of this paper is also unclear. The problem scenario seems unrealistic. To put it bluntly, one can purchase batches of semiconductor laser products from the open market to test themselves, which will produce similar results. Using FL does not change the limitation that vendors can only rely on accelerated aging tests to predict maintenance.

In terms of innovation, the paper directly uses the FedAvg framework, and the prediction and fault detection methods leveraged are also traditional.

In the threats analysis part, the paper shows three possible attacks in this specific scenario. They are probably based on the author’s assumptions, or there is no clear evidence to show the purpose of these attacks.

For example, is it possible for a model inversion attack to obtain real data from other vendors? Why not just buy a product to test directly?

During the local model poisoning attack, if an attacker tries to pollute the global model with poisoned data, the prediction results it obtains will also be meaningless, which is contrary to the fundamental motivation as the paper claimed: “Instead, each vendor receives the personalized maintenance report which contains the discrepancy between its local model and the global model, which is useful to improve the quality of products in the future.”

Lastly, in the experimental part, the paper splits data into ten pieces to simulate ten clients, which cannot reflect the diversity of products from different manufacturers.

In addition, there are some obvious language problems, as listed below:

-	In Abstract, “it is challenging to develop an accurate and reliable ML based prognostic models” -> accurate and reliable ML based prognostic models, or “an … model”

-	In Page 1, Paragraph 1, “global predictive maintenance market is expected” -> the global predictive maintenance market

-	In Page 2, Paragraph 5, “a secure aggregation protocol is tolerant to the malicious behavior of participants in a honest-majority model” -> an honest-majority model

-	In Page 2, Paragraph 5, “Such framework does not fit for our use case due to multiple reasons.” -> Such a framework

-	In Page 2, Paragraph 5, “may give negative impact on the data owner’s business.” -> give a negative impact

-	In Page 3, Paragraph 5, “whereby different semiconductor laser manufacturers (i.e vendors) collaborate …” -> (i.e., vendors)

-	In Page 5, Paragraph 4, “For example, in Fredrikson et al. (2015), authors demonstrated a model inversion attack” -> the authors

-	In Page 7, Paragraph 2, “The architecture of global model is composed of two GRU layers containing each 64 cells.” -> containing 64 cells.

**Summary Of The Paper:**

This paper designs a maintenance prediction framework for key optical network components based on federated learning. The designed framework can resist malicious environments and several kinds of attacks. The paper uses simulation data to verify that the proposed method has good predictive ability and can withstand the designed simulation attack.

**Summary Of The Review:**

In summary, although the paper tackles a novel problem and designs an FL-based framework for the target, it shows significant defects in its motivation, innovation, and experiments. As a result, I do not recommend accepting the paper.

---

### Official Review · Reviewer_JagA · 2021-11-03

**Correctness:** 3
**Technical Novelty And Significance:** 1
**Empirical Novelty And Significance:** 2
**Recommendation:** 3
**Confidence:** 4

**Main Review:**

Strengths
- This paper provides an interesting application of using ML.

Weakness
- Lack of ML technique related novelty: this paper directly leverages the existing ML techniques, eg., FedAvg and AutoEncoder, w/o proposing new techniques. I think this is just a pure ML application paper.
- Motivation doubt: I wonder whether there will be the malicious venders in the real world. If the vender wants to pollute the global model, then the malicious data/model will affect the vender's own prediction accuracy as well, then why does he/she want to do that. Not sure whether the authors can address the motivation clearer.
- Why Antoencoder is picked for anomaly detection? Why not the other approaches?

**Summary Of The Paper:**

This paper demonstrates that for the problem of doing predictive maintenance in optical networks, accurate and reliable ML models could be developed in collaborative learning without the disclosure of vendors’ data even in malicious setting. The authors did some experiments to show that 1) federated learning can help to achieve similar prediction accuracy as the centralized approach; 2) malicious behaviors may happen in federated learning, but it can be detected through autoencoder.

**Summary Of The Review:**

This paper provides an interesting application of using ML.

---

### Official Review · Reviewer_p63q · 2021-11-04

**Correctness:** 2
**Technical Novelty And Significance:** 2
**Empirical Novelty And Significance:** 3
**Recommendation:** 5
**Confidence:** 3

**Main Review:**

I think this is valuable work and I liked many parts of the paper. However, this work needs to be significantly matured prior to publication. The major issue I had with this paper is that it couldn't decide what it wanted to be. The primary benefit claim was in reducing costs of maintenance, but that was not evaluated. Instead they put extensive time and effort into discussion of security and privacy concerns, without significant ties to machine learning. They also spent extensive time reviewing material that should be assumed background reading for the ICLR audience.

Regarding the primary claim of reduced operational costs, they did not measure this directly. Rather, they evaluated a proxy measures of precision, recall, accuracy, and f1 score, which are incomplete evaluations for operational benefit. The key measure that they left out was the timeliness of detection, and this was not evaluated.

Security and privacy are always challenging to evaluate. I give the authors credit for elevating these to first class concerns in this paper. They evaluated privacy and security using robustness as a measure, with different percentages of malicious clients giving motivated "noisty" data as input. The "n-out-of-n" scheme was a nice application to limited model inversion attacks. They also made effective use of an anomaly detectors to detect data poisoning attacks. While I suspect there are ways to fool an anomaly detector, this still makes such attacks significantly more challenging.

On a more superficial note, large parts of this paper should be assumed knowledge for an ICLR audience. For example, equations 1-7 do not meaningfully improve any of their results, are not referenced again, and do not help a reader reproduce any of the results. The methods section needs more details on the structure of the training and test datasets, what are the features? how long are the series? How were your experiments setup? Much of this is in section 4.1, but I had to go looking for it.

The data that was used and baseline efforts were somewhat unsatisfying. Only one non-public dataset was evaluated, and it was unclear how well that dataset reflected many of "needs" identified in the use case - splitting a monolotic dataset into 10 parts is closer to a 10-fold cross validation scheme than a federated learning scheme. While splitting the data up this way might be required for federated learning, without clearly identfied splitting criteria (such as client 10 all were all deployed in the same geographic regon, or used hardware from the same vendor), I found it difficult to connect the experiments to the identified needs.



**Summary Of The Paper:**

This paper uses federated machine learning for predictive maintenance on optical networks. Federated learning provides a number of advantages, including security, privacy, and accuracy. The accuracy claims are motivated by having a broader set of failure examples to draw from, addressing a known issue in predictive maintenance. Privacy and security claims are important in predictive maintenance, but were less novel or well supported - they felt more like off the shelf methods applied in a new domain.

Their primary comparison was against a global model.

**Summary Of The Review:**

This is a promising, but immature work. It needs to be focused on only one of the identified problems and the evaluation needs to be significantly improved. It seems like there is potential for multiple papers in this work, but it needs to be split up and experiments carried out more carefully. This paper would also benefit from a more direct connection to the identified use case, as well as some justification of why that use case is realistic.

---

### Decision · Program_Chairs · 2022-01-20

**Decision:**

Reject

**Comment:**

The paper presents an optimization technique for optical networks based on federated learning. The motivation for using federated learning stems from the privacy of datasets arising from different operators. The performance of the method is compared to the one based on centralized learning. Despite demonstrating an interesting and promising application of a federated learning, the paper is rather weak in its methodical contribution. Its experimental evaluation, however, is rather artificial with an FL problem generated by splitting the dataset for a centralized problem into parts. No response to the reviewers' comments was provided.